

# HPO2GO: prediction of human phenotype ontology term associations for proteins using cross ontology annotation co-occurrences

Tunca Doğan

Department of Health Informatics, Graduate School of Informatics, Middle East Technical University, Ankara, Turkey

Cancer Systems Biology Laboratory (KanSiL), Graduate School of Informatics, Middle East Technical University, Ankara, Turkey

European Molecular Biology Laboratory, European Bioinformatics Institute (EMBL-EBI), Hinxton, Cambridge, UK

Corresponding author
Tunca Doğan, tdogan@metu.edu.tr, tdogan@ebi.ac.uk

## ABSTRACT

Analysing the relationships between biomolecules and the genetic diseases is a highly active area of research, where the aim is to identify the genes and their products that cause a particular disease due to functional changes originated from mutations. Biological ontologies are frequently employed in these studies, which provides researchers with extensive opportunities for knowledge discovery through computational data analysis. In this study, a novel approach is proposed for the identification of relationships between biomedical entities by automatically mapping phenotypic abnormality defining HPO terms with biomolecular function defining GO terms, where each association indicates the occurrence of the abnormality due to the loss of the biomolecular function expressed by the corresponding GO term. The proposed HPO2GO mappings were extracted by calculating the frequency of the co-annotations of the terms on the same genes/proteins, using already existing curated HPO and GO annotation sets. This was followed by the filtering of the unreliable mappings that could be observed due to chance, by statistical resampling of the co-occurrence similarity distributions. Furthermore, the biological relevance of the finalized mappings were discussed over selected cases, using the literature. The resulting HPO2GO mappings can be employed in different settings to predict and to analyse novel gene/protein—ontology term—disease relations. As an application of the proposed approach, HPO term—protein associations (i.e., HPO2protein) were predicted. In order to test the predictive performance of the method on a quantitative basis, and to compare it with the state-of-the-art, CAFA2 challenge HPO prediction target protein set was employed. The results of the benchmark indicated the potential of the proposed approach, as HPO2GO performance was among the best ($Fmax = 0.35$). The automated cross ontology mapping approach developed in this work may be extended to other ontologies as well, to identify unexplored relation patterns at the systemic level. The datasets, results and the source code of HPO2GO are available for download at: https://github.com/cansyl/HPO2GO.

## INTRODUCTION AND BACKGROUND

Systematic definition of biomedical entities (e.g., diseases, abnormalities, symptoms, traits, gene and protein attributes, activities, functions and etc.) is crucial for computational studies in biomedicine. Ontological systems, composed of standardized controlled vocabularies, are employed for this purpose. The Human Phenotype Ontology (HPO) system annotates disease records (i.e., terms and definitions about diseases together with related information) with a standardized phenotypic vocabulary (*Robinson et al., 2008*; *Köhler et al., 2016*). HPO is composed of five independent sub-ontologies namely, phenotypic abnormality (i.e., the main sub-ontology defining the basic qualities of diseases), mode of inheritance (i.e., annotates diseases in terms of mendelian or non-mendelian principles), mortality/aging (i.e., information related to age of death due to the corresponding disease), frequency (i.e., frequency of the disease in a patient cohort) and the clinical modifier (i.e., additional disease characterization such as lethality, severity, etc.). Within each sub-ontology, all terms are related to each other with a parent–child relationship, where each child term defines a specific aspect of its parent. HPO has a directed acyclic graph (DAG) structure. The sources of the disease information in HPO are Orphanet (*Rath et al., 2012*), DECIPHER (*Firth et al., 2009*), and OMIM (*Amberger et al., 2014*) databases. Each term in the phenotypic abnormality sub-ontology define a specific type of abnormality encountered in human diseases (e.g., HP:0001631 - atrial septal defect). The generation of HPO terms (and their associations with diseases) are carried out via both manual curation efforts and automated procedures (e.g., text mining). The curation job is usually done by experts by reviewing the relevant literature publications along with the disease centric information at various biomedical data resources. For each association between a disease term and an HPO term, there is an evidence code tag to specify the source of the information (i.e., curated or automated). The evidence codes used in HPO are IEA (inferred from electronical annotation), PCS (published clinical study), ICE (individual clinical experience), ITM (inferred by text mining), TAS (traceable author statement). As of January 2018, the growing library of HPO contains nearly 12,000 phenotype terms, providing more than 123,000 annotations to 7,000 different rare (mostly Mendelian) diseases and the newly added 132,000 annotations to 3,145 common diseases (*Groza et al., 2015*). A long-term goal of the HPO project is for the system to be adopted for clinical diagnostics. This will both provide a standardized approach to medical diagnostics and present structured machine readable biomedical data for the development of novel computational methods. Apart from phenotype-disease associations, which is the main aim of the HPO project, HPO also provides phenotype-gene associations by using the known rare disease—gene relations (i.e., the information which is in the form of: "certain mutation(s) in *Gene X* causes the hereditary *Disease Y*"), directly using the abovementioned disease centric resources (e.g., Orphanet and OMIM). The disease-gene associations in

the source databases are produced by expert curation from the publications of clinical molecular studies. The associations between HPO terms and biomolecules, together with the downstream analysis of these associations, help in disease gene identification and prioritization (*Köhler et al., 2009*). With the mapping of phenotypes to human genes, HPO currently (January 2018) provides 122,166 annotations between 3,698 human genes and 6,729 HPO terms.

The Gene Ontology (GO) is an ontological system to define gene/protein attributes with an extensive controlled vocabulary (*Gene Ontology Consortium, 2014*). Each GO term defines a unique aspect of biomolecular attributes. Similar to other ontological systems, GO has a directed acyclic graph (DAG) structure, where terms are related to each other mostly with ''is_a'' or ''part_of'' relationships. GO is composed of three categories (i.e., aspects) in terms of the type of the defined gene product/protein attribute such as: (i) molecular function—MF (i.e., the fundamental function of the protein at the molecular level; e.g., GO:0016887—ATPase activity), (ii) biological process—BP (i.e., the high level process, in which the protein plays a role; e.g., GO:0005975—carbohydrate metabolic process), and (iii) cellular component—CC (i.e., subcellular location, where the protein carries out its intended activity; e.g., GO:0016020—membrane). Similar to the other ontological systems, the basic way of annotating a gene or protein with a GO term is the manual curation by reviewing the relevant literature. GO also employs the concept of ''evidence codes'', where all annotations are labelled with descriptions indicating the quality of the source information used for the annotation (e.g., ECO:0000006 - experimental evidence, ECO:0000501 – IEA: evidence used in automatic assertion). UniProt-GOA (Gene Ontology Annotation) database (*Huntley et al., 2015*) houses an extensive collection of GO annotations for UniProt protein sequence and annotation knowledgebase records. In the UniProtKB/Swiss-Prot database (i.e., housing manually reviewed protein entries with highly reliable annotation) version 2018_02, there are a total of 2,850,015 GO term annotations for 529,941 protein records; whereas in UniProtKB/TrEMBL database (i.e., housing mostly electronically translated uncharacterized protein entries) version 2018_02, there are a total of 189,560,296 GO term annotations for 67,760,658 protein records. Most of the annotations for the UniProtKB/TrEMBL database entries are produced by automated predictions (*UniProt Consortium, 2017*).

Due to the high volume of experimental research that (i) discover new associations between biomolecules and ontological terms, and (ii) produce completely new and uncharacterized gene/protein sequences, curation efforts are having difficulty in keeping up with the annotation process. To aid manual curation efforts, automated computational methods come into play. These computational methods exploit the approaches and techniques widely used in the fields of data mining, machine learning and statistics to produce probabilistic associations between biomedical entities. The Critical Assessment of Functional Annotation (CAFA) challenge (*Radivojac et al., 2013*; *Jiang et al., 2016*) aims to evaluate the automated methods that produce GO and HPO term association predictions for protein entries, on standard temporal hold-out benchmarking datasets. Now after its third instalment, CAFA organization has already brought together a research community,
dedicated to elevate the capabilities of automated function prediction approaches closer to the level of expert review.

Protein function prediction using GO terms is a very active area of research where various types of approaches utilizing: amino acid sequence similarities (*Hawkins et al., 2009*), 3D structure analysis (*Roy, Yang & Zhang, 2012*), semantic similarities between the ontological terms (*Falda et al., 2012*), gene expression profiles (*Lan et al., 2013*), protein–protein interactions—PPIs (*Wass, Barton & Sternberg, 2012*), shared functional domains and their arrangements (*Fang & Gough, 2012*; *Finn et al., 2016*; *Doğan et al., 2016*) and ensemble approaches that exploit multiple feature types (*Wass, Barton & Sternberg, 2012*; *Cozzetto et al., 2013*; *Lan et al., 2013*; *Rifaioglu et al., 2018*) are employed to model the proteins and to transfer the functional annotations from characterized proteins (i.e., the ones that have reliable annotation) to the uncharacterized ones with highly similar features. Known GO associations of genes and proteins are also used in different contexts in the literature. For example, the method "MedSim" uses the semantic similarities between GO terms for the prioritization of disease genes (*Schlicker, Lengauer & Albrecht, 2010*). The method "spgk" uses a shortest-path graph kernel to compute functional similarities between gene products using their GO annotations and the term relations on the GO DAG (*Alvarez, Qi & Yan, 2011*).

The automated prediction of the associations between human genes/proteins and phenotype/disease defining ontological terms is also a non-trivial task. The resulting predictions can then be utilized to identify large-scale novel disease-gene-pathway/system relations. The identification of direct disease-gene relations is a widely studied topic (*Moreau & Tranchevent, 2012*). A considerable amount of the existing literature about disease-gene associations involve the calculation of semantic similarities between gene products, based on the already existing ontological term annotations (*Washington et al., 2009*; *Smedley et al., 2013*; *Deng et al., 2015*; *Rodríguez-García et al., 2017*). For example, the method "PhenomeNET" was employed to generate mappings between the highly related terms across similar ontological systems (*Rodríguez-García et al., 2017*) such as the HPO, Mammalian Phenotype Ontology—MP (*Smith, Goldsmith & Eppig, 2005*), Human Disease Ontology—DO (*Kibbe et al., 2014*) and Orphanet Rare Disease Ontology—ORDO (*Vasant et al., 2014*) for discovering novel gene-disease associations. However, semantic similarity based approaches sometimes suffers from the low coverage of the HPO annotations on the protein space. The authors of two recent studies have investigated this issue (*Kulmanov & Hoehndorf, 2017*; *Peng, Li & Shang, 2017*). In this context, increasing the coverage of HPO annotations by predicting gene/protein-HPO term associations may help semantic similarity based association studies.

There are only a few examples of HPO term-protein association prediction methods in the literature. In the "dcGO" method, the authors mapped ontological terms (including HPO) to protein domains, which are the functional units, and transferred the ontology mapping to proteins according to known domain annotations (*Fang & Gough, 2012*). The objective in the "PHENOstruct" method is the prediction of gene-HPO term associations using heterogeneous biological data consisting of protein–protein interactions (PPIs), GO annotations, literature relations, variants and known HPO annotations, together
with a structured SVM classifier (*Kahanda et al., 2015*). In this sense, it is interesting see another method that utilize already existing GO annotations to predict HPO terms. Nevertheless, according to the authors, the contribution of the GO features to the overall predictive performance of PHENOstruct was minimal. A text mining based method "EVEX", was employed for protein-HPO term association prediction. Originally, EVEX utilizes text mining approaches for large-scale integration of heterogeneous biological data and event extraction to generate a structured resource of relations, to be used in pathway curation (*Van Landeghem et al., 2013*). In the context of HPO term prediction, EVEX scans the literature to detect proteins and phenotypic terms that co-occur on the same text corpus, and associates them with each other based on certain criteria. According to the CAFA2 challenge results (*Jiang et al., 2016*), the participating method EVEX was the top performer. A network based HPO prediction method was the "RANKS", in which the authors developed a flexible algorithmic scheme for heterogeneous biological network analysis, and used previously generated functional Interaction and functional human gene networks for gene-HPO term association prediction (*Valentini et al., 2016*). In a recent study, the authors proposed two hierarchical ensemble methods: (i) the Hierarchical Top-Down, and (ii) the True Path Rule, for gene-HPO term associations; in which the hierarchical graph structure of HPO has been utilized together with the RANKS algorithm and the SVM classifier (*Notaro et al., 2017*).

The text mining approach is highly effective for predicting gene-disease relations in disease gene prioritization studies (*Krallinger, Valencia & Hirschman, 2008*). However, this approach suffers from low coverage in some cases, due to knowledge limitation in the literature. If a certain abnormality and a gene/protein has not been studied together in the same framework yet, it is often not possible to identify the relation. Network based methods are proposed on top of either text-mining results, protein–protein interactions and/or pathway data (*Bromberg, 2013*; *Guney & Oliva, 2014*; *Guala & Sonnhammer, 2017*) to detect indirect relations, which greatly increased the coverage; nevertheless, they still moderately rely on the previously reported relations. It is also important to note that any predictive approach is limited by the quality and the coverage of its source information. However, the predictive output of different approaches often complement each other, contributing to fill different portions of the missing information in the knowledge space. Due to this reason, developing novel approaches to complement the conventional methods is crucial for the automated ontological association prediction. The observed low performance of even the best methods in the HPO term prediction track of the CAFA2 challenge displayed the necessity of novel approaches for the biomedical entity relation prediction.

In this study, a new approach "HPO2GO" is proposed to produce phenotypic abnormality HPO term associations to both GO terms and human genes/proteins, with the analysis of co-annotation fractions between the HPO and GO term combinations. For this, HPO and GO terms that are continually co-occurring on different genes/proteins as annotations are linked to each other (i.e., the system training step), entitled the "HPO2GO mappings". After that, proteins with a linked GO term annotation receives the corresponding HPO term as the phenotypic term prediction (i.e., the application step), entitled the "HPO2protein predictions". The idea here is to associate a HPO term $Y$ with a

GO term $X$ in the sense that: "if a protein loses its function defined by the GO term $X$ (or at least a reduction in the defined functionality) as a result of a genetic mutation, the loss of function may cause the disease, which is defined by the phenotype term $Y$". This idea is based on the nature of annotating genes/proteins with HPO terms; for example, only the functionally perturbed variants of these genes/proteins (e.g., disease causing variants) are associated with the relevant genetic diseases and their defining phenotypic abnormality terms. Mutations often lead to diseases by causing either a loss of the existing functionality or a gain of new functionality in the gene products. As a result, if the HPO term $Y$ and the GO term $X$ are observed to be frequently co-occurring on different proteins, then the lost function, which gave way to the corresponding disease may be the one defined by the GO term $X$. This logic would make biological sense especially when the corresponding function is a large-scale biological process. This approach exploits the significantly higher coverage of GO term annotations for genes/proteins, compared to the HPO term annotations; to produce novel gene/protein - HPO term associations.

In order to test the biological relevance of this approach, selected HPO2GO mappings were manually examined. Additionally, the proposed methodology was employed to predict HPO terms for the human protein target dataset provided in the CAFA2 challenge. Using the benchmark set, the prediction performance was calculated and compared with the state-of-the-art HPO prediction methods. Another set of HPO2GO mappings were generated for this test, using the temporal hold-out training data provided in CAFA2. Finally, the up-to-date HPO2GO mappings were employed to generate HPO term predictions to human protein entries in the UniProtKB/Swiss-Prot database (i.e., HPO2protein predictions). The training and test datasets, along with the source code of the proposed methodology and the analyses are available for download at https://github.com/cansyl/HPO2GO.

## METHODS

### Dataset construction

To generate the training sets: first, gene to HPO term association file was downloaded from the HPO web-site (January 2017 version of the file named: "ALL_SOURCES_ALL_FREQUENCIES_ genes_to_phenotype.txt"). This file contained 153,575 annotations between 3,526 human genes and 6,018 HPO terms. This file is shared in the HPO2GO repository with the filename: "HPO_gene_to_phenotype_annotation_01 _2017_ALL_SOURCES_ALL_FREQUENCIES.txt". These gene-phenotype associations are generated as a part of the HPO project by merging two source files: first, the gene-disease associations provided in the incorporated disease centric databases (e.g., OMIM and Orphanet); and second, the disease-phenotype associations generated by HPO. In HPO, "genes_to_phenotype" file only contains the asserted (i.e., specific) annotations to genes; whereas "phenotype_to_genes" file contains all annotations propagated through the root of the HPO DAG, according to the true path rule. As a result, parents of the asserted terms are included as well. In this study, the asserted annotations are used (in terms of HPO) at the input level in order to avoid the propagation of potential false positive annotations that may be presented in the source dataset. This application usually comes with the cost

of increased number of false negatives; however, avoiding the potential false positives is considered more important here.

In order to generate the second training dataset, all GO term annotations to the human proteins in UniProtKB with the manually assigned (curator assigned) evidence codes (i.e., EXP, IDA, IPI, IMP, IGI, IEP, IBA, IC, IKR, ISS, NAS, ND and TAS) were downloaded from the UniProt-GOA database 2017_01 version, using the QuickGO browser (filename: "GOA_UniProt_human_protein_annotation.tsv"). The reason behind not using electronically assigned annotations (i.e., evidence code: IEA) was that these annotations have a reduced level of annotation reliability compared to the curator assigned ones; as a result, they may contain erroneous cases (i.e., false positives). After eliminating the repeating (i.e., redundant) annotations, the finalized file contained 179,651 GO annotations between 18,577 unique human genes and 14,632 GO terms (filename of the finalized GO annotation file: "GO_annot_human_proteins_UniProtGOA_01_2017.txt"). An additional column containing the corresponding HGNC symbols (i.e., gene symbols) of the coding genes was also included in the downloaded GO annotation file. This column was later used to combine the GO annotations with the HPO annotations, since the HPO annotation file includes the gene symbols.

## Applied methodology

The proposed methodology is divided into two steps: (i) training of the system (i.e., the generation of the HPO2GO mappings), and (ii) the application step (i.e., the prediction of HPO term-protein associations – HPO2protein, using the previously generated HPO2GO mappings).

Figure 1 represents the complete HPO2GO mapping (i.e., training) procedure. For the training of the system, first, the HPO and GO annotation datasets were prepared (Figs. 1A and 1B) and the initial HPO-GO mappings were generated (Fig. 1C) by identifying the genes/proteins shared between individual HPO and GO terms (i.e., the cases where HPO and GO terms were co-annotated to the same genes/proteins). This mapping has generated 1,433,208 unique pairs between 6,005 HPO terms and 9,685 GO terms. At this point, it was observed that some of GO and HPO terms were annotated to high number of proteins, and it was highly probable for them to co-occur on the same protein once or twice just by chance. In order to eliminate these cases, a filtering procedure was required. For each HPO-GO term pair, a co-occurrence similarity measure, inspired from an information-theoretic definition of similarity (*Lin, 1998*), has been calculated. The co-occurrence similarity formulation is given in Eq. (1).

$$S_{HPOi, \; GOj} = \frac{2 * N_{G \; HPOi \& GOj}}{N_{G \; HPOi} + N_{G \; GOj}}. \tag{1}$$

Here, $S_{H \; POi,GOj}$ is the co-occurrence similarity between the HPO term "$HPOi$" and the GO term "$GOj$", $N_{G \; HPOi \& GOj}$ is the number of genes/proteins where these terms are annotated together, $N_{G \; HPOi}$ is the total number of genes with the annotation "$HPOi$", and $N_{G \; GOi}$ is the total number of genes with the annotation "$GOi$".

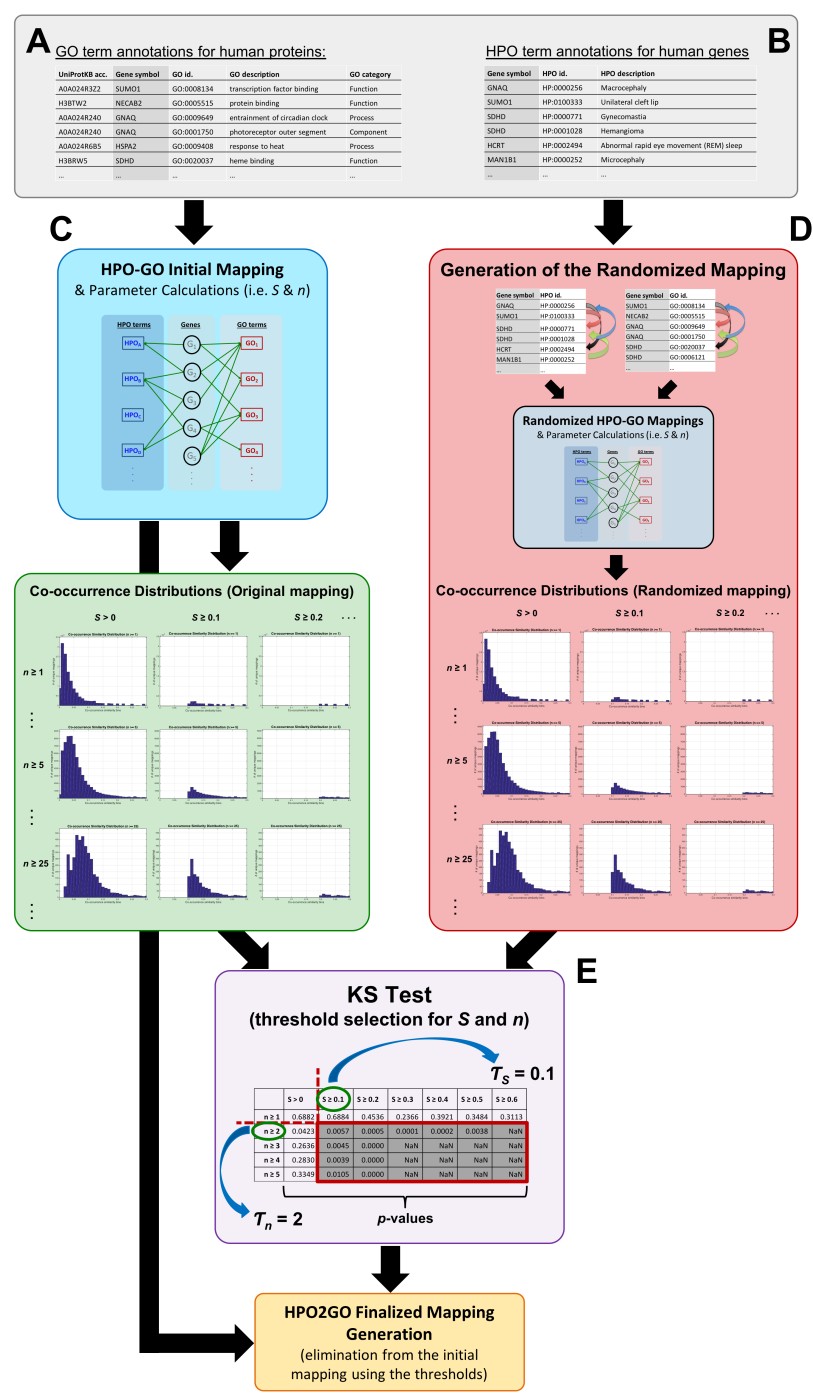

**Figure 1** **Schematic representation of the whole HPO2GO mapping (i.e., training) procedure.** (A) The source GO annotation dataset; (B) the source HPO annotation dataset; (C) initial mapping of the HPO and GO terms together with the parameter calculation and plotting the co-occurrence similarity distributions; (D) generation of the randomised annotation files and their mapping, and co-occurrence similarity distribution plotting for the random mappings; (E) statistical resampling (via KS-test) of the mappings using the co-occurrence similarity distributions, selection of the thresholds, generation of the finalised HPO2GO mappings with the elimination of unreliable mappings considering the selected threshold values.

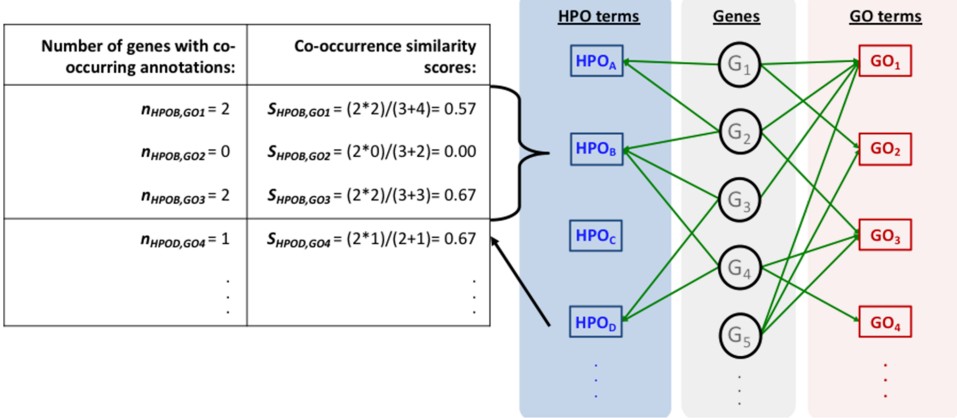

**Figure 2** **Representation of the initial HPO-GO mapping process.** Representation of the initial HPO-GO mapping process together with the calculation of co-occurrence similarities ($S$) and the number of genes with co-occurring annotations ($n$), on a toy example.

The mapping process and the co-occurrence similarity calculation are shown in Fig. 2 with a toy example. Following the calculation of the co-occurrence similarities between all HPO-GO pairs, a thresholding operation was applied in order to distinguish between relevant mappings and the random ones. Two parameters were used for the thresholding operation: (i) the co-occurrence similarities: $S$, and (ii) the number of genes with co-occurring annotations: $n$. The aim behind employing a second parameter (i.e., $n$) was to eliminate the potential random pairing cases, where the co-occurrence similarity is still high. These cases are rare; however, it is still possible to observe a few of them especially when $n$ is very small, due to extremely high number of term combinations. In Fig. 2, this situation is represented on the toy example, here $S_{HPOD,GO4}$ is equal to $S_{HPOB,GO3}$; however the $HPO_D$-$GO_4$ mapping is probably less reliable compared to $HPO_B$-$GO_3$ since $n_{HPOD,GO4}$ is equal to 1.

Statistical resampling was used to determine the optimal parameter values (to be used as thresholds), that separate meaningful mappings from random ones. A permutation (i.e., randomization) test was constructed for this purpose. At this point, a randomized HPO-GO term mapping table was required, which was generated (Fig. 1D) by first, shuffling the indices of the original "HPO vs. gene" and "GO vs. gene" annotation tables; and second, calculating both the randomized co-occurrence similarities (i.e., $S_R$) and the number of genes with co-occurring annotations (i.e., $n_R$) for each random HPO-GO mapping. For each arbitrarily selected $S$ (i.e., $S > 0$, $S \geq 0.1$, $S \geq 0.2$, …, $S \geq 0.6$) and $n$ (i.e., $n \geq 1$, $n \geq 2$, …, $n \geq 5$) threshold value combination, the original GO-HPO mappings with lower than the threshold $S$ and $n$ values were deleted and a co-occurrence similarity distribution histogram was plotted using the remaining mappings (i.e., histograms plots in Fig. 1 and in Fig. 3). The same procedure was applied for the randomized mapping set as well. Finally, the Kolmogorov–Smirnov test (KS test) (*Lilliefors, 1967*; *Hollander, Wolfe & Chicken, 2013*) is employed to calculate a test statistic for estimating whether the samples from the random
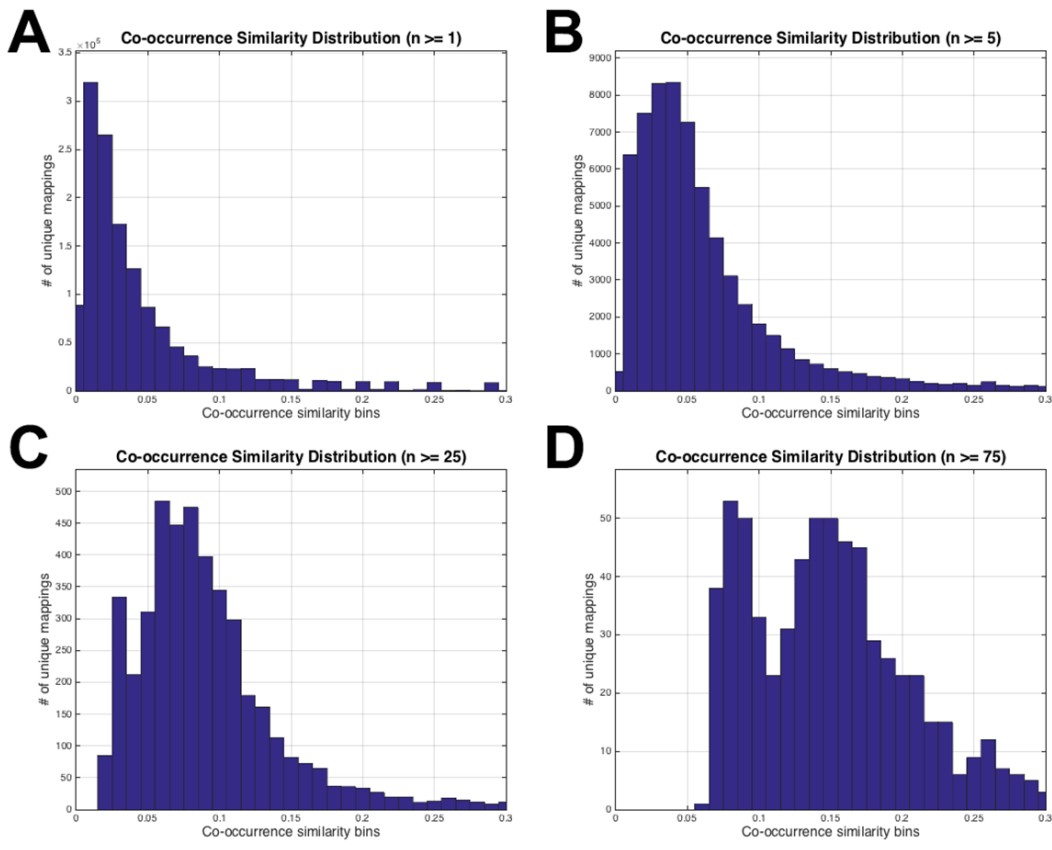

**Figure 3** **HPO-GO initial mappings co-occurrence similarity distributions.** Each plot is drawn for a different value of the number of co-annotated genes (i.e., $n$): (A) $n \geq 1$; (B) $n \geq 5$; (C) $n \geq 25$; (D) $n \geq 75$.

and the original sets (at each $S$ and $n$ selection) are from the same distribution or not. KS is a nonparametric test of 1-dimensional probability distributions that can be used to compare two samples, considering the quantized distance between the samples. The null hypothesis states that the two samples are drawn from the same distribution. Here, the distribution (i.e., histogram) of the $S$ values for the original and the randomized mapping sets represent the two samples. The reason behind using histograms instead of the actual $S$ values was that, both high and low $S$ values were presented in both distributions; as a result, the significance test by checking sample distances approach would not work. However, the frequencies of these high and low $S$ values are different from each other in the original and the random distributions. If the null hypothesis is accepted at a selected threshold value pair ($S$ and $n$), which means that the distributions are not statistically different from each other, then it is concluded that the selected thresholds failed to eliminate the random pairings in the original mapping (i.e., a higher threshold is required). The lowest threshold values, where the samples from the two distributions became significantly different from each other, were selected as the official thresholds. Excessive threshold values were not considered in order not to eliminate too many GO-HPO mappings. After the determination of the parameter

values (i.e., $S$ and $n$ thresholds), the HPO2GO mappings were finalized, which ended the training process.

An alternative to the statistical resampling methodology used here would be employing a direct approach such as an exact test (e.g., Fisher's exact test), and to calculate a significance value for each HPO-GO mapping individually. The advantage of this approach would be accepting or rejecting the null hypotheses specific to each mapping (as opposed to finding a global significance using the whole data at once, as applied in this study). The main disadvantage of this approach would be the diminished statistical power to calculate significance values for the mappings where the number of instances are too low. Unfortunately, this is the case for most of the HPO-GO mappings as the mean number of occurrences (for all HPO-GO mappings) in the original or in the random sets were 1.77 and 1.48, respectively. As a result, the direct approach was not considered in this study.

HPO2protein prediction step was a simple procedure, where query proteins were annotated with the HPO terms, by taking their already existing GO annotations into account. HPO2GO mappings were employed for this purpose. There were a total of three application runs in this study using: (i) CAFA2 targets as the query set (for the performance tests and for the comparison with the state-of-the-art), (ii) CAFA3 targets as the query set (to officially participate in the CAFA3 challenge), and (iii) all human protein entries in the UniProtKB/Swiss-Prot database (to generate the HPO2protein predictions).

## Performance evaluation

In this study, it was not possible to use a standard fold based cross-validation to measure the performance and to determine the parameter values in the training procedure, since in most cases, the number of genes/proteins that have a co-occurring HPO-GO term annotations were extremely low. As a result, it was not impossible to separate the samples into training and validation sets. Instead, the optimal parameter values were determined by using statistical resampling. However, a performance test was still required in order to assess the success of the proposed approach. For this, CAFA2 challenge benchmark set was employed. Due to the fact that CAFA2 challenge was long before the analysis done in this study, HPO2GO mappings were re-generated using both the GO and HPO annotation data from January 2014. This was followed by the production of the HPO-protein association predictions on the CAFA2 target gene set. This analysis both served as a performance test with the temporal hold-out data (one of the hardest and most informative tests for predictive models) and a performance comparison with the state-of-the-art (i.e., other HPO prediction methods participated in CAFA2). The most basic definitions of the evaluation metrics used in this test; *recall*, *precision*, *Fmax* and minimum semantic distance (*Smin*) are shown in Eqs. (2)–(5).

$$Rc_{\tau i} = \frac{TP_{\tau i}}{TP_{\tau i} + FN_{\tau i}} \tag{2}$$

$$Pr_{\tau i} = \frac{TP_{\tau i}}{TP_{\tau i} + FP_{\tau i}} \tag{3}$$

$$F_{max} = \max_{i=1\ldots N} \left\{ \frac{2 * Pr_{\tau i} * Rc_{\tau i}}{Pr_{\tau i} + Rc_{\tau i}} \right\} \tag{4}$$

$$S_{min} = \min_{i=1\ldots N} \left\{ \sqrt{Ru_{\tau i}^2 + Mi_{\tau i}^2} \right\}. \tag{5}$$

In Eqs. (2)–(4); $TP_{\tau i}$, $FN_{\tau i}$, $FP_{\tau i}$, $Rc_{\tau i}$ and $Pr_{\tau i}$ represent the number of true positives, the number of false negatives, the number of false positives, *recall* and *precision* values, respectively; at the $i$ th probabilistic score threshold ($\tau_i$). *Fmax* correspond to the maximum of the *F-score* values (i.e., harmonic mean of *precision* and *recall*, shown inside the curly brackets in Eq. (4)) calculated for each arbitrarily selected probabilistic score threshold. $i = 1\ldots N$ represents there are $N$ different arbitrarily selected probabilistic score thresholds. Higher *Fmax* values indicate higher performance. $Ru_{\tau i}$ and $Mi_{\tau i}$ in Eq. (5) corresponds to remaining uncertainty and normalized misinformation at the $i$th probabilistic score threshold ($\tau_i$), respectively. *Smin* is the minimum semantic distance. Lower *Smin* indicate higher performance. Also, a weighted version of the *Fmax* measure has been calculated. Weighting procedure was applied using the information content of each ontology term; so that, more informative terms obtained higher weights. Information regarding the calculation of remaining uncertainty, misinformation and term based information contents can be found in (*Clark & Radivojac, 2013*) and in (*Jiang et al., 2016*).

In the proposed method, probabilistic scores for each HPO-protein association prediction is calculated using the term co-occurrence similarity scores in Eq. (1). If the mapping between the terms *HPOi* and *GOj* received the co-occurrence similarity score $S_{HPOi,GOj}$, then all proteins that receive the *HPOi* prediction due to the presence of *GOj* annotation obtain the probabilistic prediction score: $S_{HPOi,GOj}$. The calculation of the score in Eq. (1) is set to range between 0 and 1; as a result, it can directly be used as a probabilistic score. Apart from that, probabilistic score thresholds represent values, under which the predictions are discarded. This way, a different set of predictions are given for each arbitrarily selected probabilistic score threshold, leading to different precision and recall values. It is important to note that, probabilistic score thresholds are different from the thresholds we used to filter out unreliable HPO2GO mappings during the training process. The probabilistic score thresholds are used here (i.e., after the production of HPO2protein predictions) to produce binary predictions from continuous prediction scores, to be able to calculate performances. More details regarding the CAFA2 evaluation metrics are given in (*Jiang et al., 2016*).

## RESULTS

### Statistical analysis of the mappings

The initial HPO to GO mappings were generated according to the procedure explained in the 'Methods' section (Fig. 2). The initial mapping of the original set resulted in 1,433,208 mappings between 6,005 HPO terms and 9,685 GO terms. The same procedure
**Table 1** Statistics of the initial (i.e., raw) original and randomized HPO-GO mappings ($n \geq 1$).

| $S$ | # of mappings | | # of mapped HPO terms | | # of mapped GO terms | |
|---|---|---|---|---|---|---|
| | Original mapping | Random mapping | Original mapping | Random mapping | Original mapping | Random mapping |
| $= 1$ | 2,433 | 1,898 | 844 | 877 | 1,108 | 1,265 |
| $\geq 0.9$ | 2,440 | 1,898 | 848 | 877 | 1,109 | 1,265 |
| $\geq 0.8$ | 2,658 | 1,899 | 962 | 878 | 1,179 | 1,266 |
| $\geq 0.7$ | 2,805 | 1,899 | 1,028 | 878 | 1,212 | 1,266 |
| $\geq 0.6$ | 7,355 | 5,249 | 1,941 | 1,653 | 2,577 | 2,844 |
| $\geq 0.5$ | 8,075 | 5,252 | 2,188 | 1,655 | 2,712 | 2,847 |
| $\geq 0.4$ | 15,462 | 9,724 | 3 014 | 2,243 | 4,053 | 4,207 |
| $\geq 0.3$ | 32,393 | 21,615 | 4,082 | 3,017 | 6,011 | 6,081 |
| $\geq 0.2$ | 63,439 | 43,593 | 5,032 | 3,662 | 7,569 | 7,490 |
| $\geq 0.1$ | 181,048 | 134,038 | 5,920 | 5,199 | 8,884 | 9,005 |
| $> 0.0$ | 1,433,208 | 1,543,917 | 6,005 | 5,995 | 9,685 | 9,685 |

for the randomized set produced 1,543,917 mappings between 5,995 HPO terms and 9,685 GO terms. The initial HPO-GO mappings for both the original and the randomized sets are available for download in the repository of the study (respective filenames: "HPO_GO_Raw_Original_Mapping.txt" and "HPO_GO_Random_Mapping.txt"). It was expected that the mappings generated from the random set would have lower co-occurrence similarity values on average compared to the original set mappings; in other words, they would contain less number of mappings for a particular co-occurrence similarity value. Table 1 displays the comparison of the number of mappings for different co-occurrence similarity values, between the original and the randomized sets. As observed from Table 1, when $S > 0$ there is no significant difference between the mappings; however as $S$ is increased, the difference between the mappings becomes clear. Also, when $S$ is increased, the number of mapped HPO and GO terms were decreased since many terms did not have any mappings that satisfied the stringent $S$ values. The parameter $n$ was not taken into account while calculating the statistics in Table 1 (i.e., $n \geq 1$ for all values in the table).

The histograms in Fig. 3 display the co-occurrence similarity distributions (i.e., $S$) for arbitrarily selected $n$ values. As observed from the histograms, when the mappings with low $n$ values are eliminated, the distributions shift to the right (i.e., the mean of $S$ increases), which can be interpreted as the mappings became more reliable. However, excessive values of $n$ thresholds leave only a few mappings to work with, especially at $n = 25$ and $n = 75$ (please refer to the number of mappings at the vertical axis of Figs. 3C and 3D). Histograms in Fig. 3 also show that thresholding the mappings using only $n$ (not using $S$ at all) would not be sufficient because there are mappings with very low $S$ values even at very high $n$ thresholds (i.e., 25 and 75). This observation verified the decision to use both of the parameters for the filtering operation. At this point, the statistical resampling (i.e., KS test) was applied since it was not possible to determine the optimal $n$ threshold by just manually checking the histograms.

**Table 2 KS test significance values for the comparison of original vs. randomized distributions at different co-occurrence similarity (S) and the number of co-annotated genes (n) thresholds.**

| KS test statistic | | Co-occurrence similarity threshold | | | | | | |
|---|---|---|---|---|---|---|---|---|
| | | $S > 0$ | $S \geq 0.1$ | $S \geq 0.2$ | $S \geq 0.3$ | $S \geq 0.4$ | $S \geq 0.5$ | $S \geq 0.6$ |
| | $n \geq 1$ | 0.6882 | 0.6884 | 0.4536 | 0.2366 | 0.3921 | 0.3484 | 0.3113 |
| | $n \geq 2$ | 0.0423 | **0.0057** | 0.0005 | 0.0001 | 0.0002 | 0.0038 | NaN |
| # of co-annotated genes threshold | $n \geq 3$ | 0.2636 | 0.0045 | 0.0000 | NaN | NaN | NaN | NaN |
| | $n \geq 4$ | 0.2830 | 0.0039 | 0.0000 | NaN | NaN | NaN | NaN |
| | $n \geq 5$ | 0.3349 | 0.0105 | 0.0000 | NaN | NaN | NaN | NaN |

In order to find the minimum $S$ and $n$ values that significantly separate the original mapping from the randomized mapping, 35 different distributions, all combinations of the selected $n$ (i.e., $n \geq 1, 2, \ldots, 5$) and $S$ (i.e., $S > 0$, $S \geq 0.1, \ldots, 0.6$) values, were prepared and tested individually against the co-occurrence distribution of the random mapping, generated with the same $S$ and $n$ thresholds. This test resulted in 35 different $p$-value calculations and the minimum parameter values that satisfied the statistical significance (i.e., rejection of the null hypothesis, which states that the two samples are from the same distribution) were selected. Table 2 displays the significance results of all KS tests. The cells with "NaN" indicate the cases, where the test could not be completed due insufficient number of samples to calculate the statistic. The incomplete tests did not constitute problem since the aim here was observing the minimum threshold values, where the distributions significantly diverge from each other (NaNs are located far away from this point). In Table 2, the cell with the $p$-value written in bold font (i.e., ~0.0057) signifies the point, where the corresponding thresholds $n \geq 2$ and $S \geq 0.1$ yielded the required significance ($p$-value < 0.01); and thus, these values were selected as the finalized thresholds. This means that, all of the mappings with $n < 2$ and $S < 0.1$ were considered unreliable and eliminated from the initial HPO-GO mappings.

Figure 4 displays the total number of unique mappings (vertical axis) with co-occurrence similarity values greater than the corresponding threshold value (horizontal axis), for the original and the randomized distributions on the blue and red coloured curves, respectively. Figure 4A shows the plot for the combination with greater than or equal to one co-annotated gene (i.e., $n \geq 1$), Fig. 4B displays the same value for $n \geq 2$, Fig. 4C and D for $n \geq 3$ and 4; respectively. The differences between Figs. 3 and 4 is that: (i) in Fig. 4 cumulative number of mappings are given (i.e., all mappings left after thresholding with $S \geq 0.1, 0.2, \ldots$), whereas in Fig. 3, the number of mappings that fall into each $S$ bin is given; and (ii) in Fig. 4, plots are given for $n \geq 1, 2, 3$ and 4 since the aim was to display the curves around the selected threshold $n$ value; whereas in Fig. 3, there are plots for $n \geq 1, 5, 25$ and 75 to visually indicate the distribution shifts especially at high $n$ values (i.e., $n = 25$ and $n = 75$). Figure 4 was drawn as a visual representation of the likeness between the original and the randomized distributions at different parameter selections. As observed from Fig. 4, the distributions diverged from each other at $n \geq 2$, which also is consistent with the KS test results. Considering the co-occurrence similarity parameter, $S \geq 0.1$ produced a clear separation between the original and the randomized distributions as long as $n$ is greater
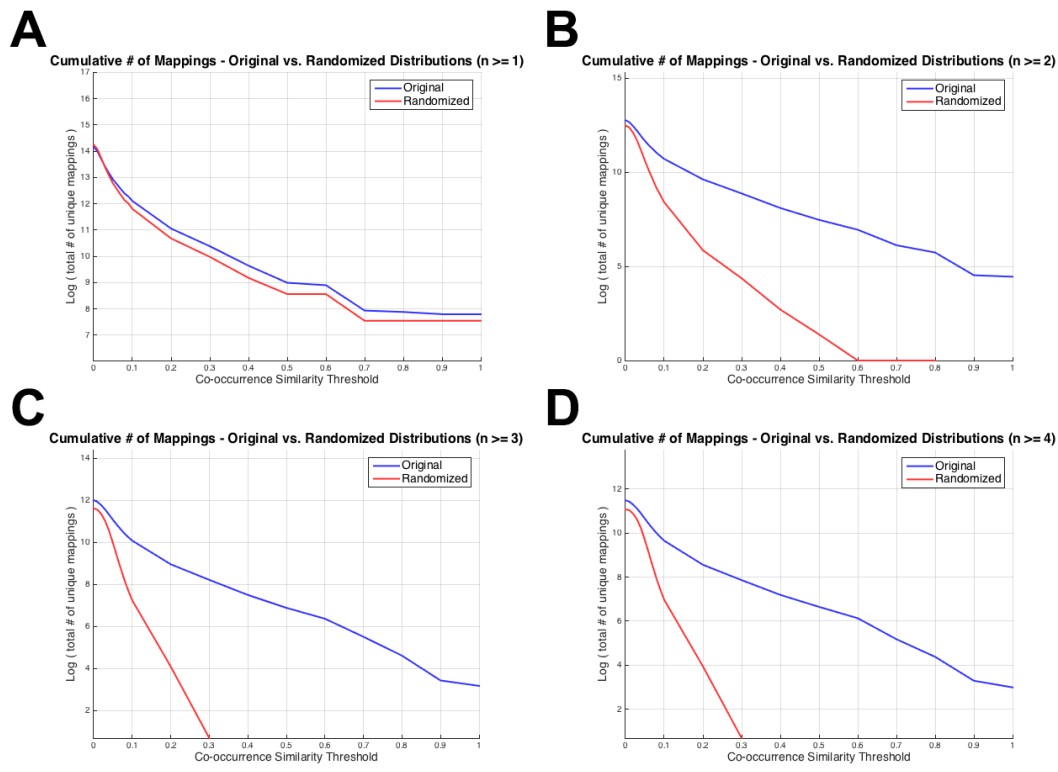

**Figure 4  Cumulative plots displaying the number of HPO-GO mappings for the original (blue curve) and the randomized (red curve) distributions.** Horizontal axis displays the arbitrarily selected co-occurrence similarity thresholds (i.e., $\tau_S$), and the vertical axis represents the logarithm of the total number of mappings left after the application of the corresponding threshold. Each plot is drawn for a different value of the number of co-annotated genes (i.e., $n$). As the threshold (i.e., the minimum required co-occurrence similarity value to keep a mapping in the system) increases more mappings are eliminated; thus, a monotonic decrease was observed for all plots.

than 1. Following the HPO-GO mapping elimination according to the selected thresholds, finalized HPO2GO mappings contained 45,805 associations between 3,693 HPO terms and 2,801 GO terms. HPO2GO mappings are available for download in the repository of the study (filename: "HPO2GO_Finalized_Mapping.txt").

It was only possible to use a small portion of the input GO annotations for the generation of the HPO2GO mappings because the number of HPO annotated genes were only 3,526; whereas, the number of GO annotated human genes were 18,577. Since mappings can be done over the genes/proteins with co-occurring GO and HPO annotations, only 3,526 genes/proteins were used in the process. The remaining 15,051 human genes with GO annotations were only used in the application step (i.e., HPO2protein), to predict HPO term associations.

## Comparison of HPO2GO with the manual HPO-GO associations

As a part of the main HPO project, a sub-set of the HPO terms has been mapped to the relevant terms from different ontologies (e.g., anatomy, Gene Ontology process or cell type) to yield semantic interoperability with these systems. Finalized HPO2GO
mappings were compared with these manual GO associations in order to observe the correspondence. The manual mappings are available in the "hp.owl" ontology file in the HPO repository. From the most up to date version (03_2018) of the owl file, the corresponding associations have been extracted and stored in the HPO2GO repository with the filename: "HPO_manual_GO_associations_03_2018.txt". There were 489 manual associations between 488 HPO terms and 239 GO terms, which was significantly lower compared to the HPO2GO mappings (i.e., 45,805 mappings between 3,693 HPO terms and 2,801 GO terms). Considering the most frequent GO terms in these manual associations, 71 HPO terms were associated with the GO term "ossification" (GO:0001503), 51 HPO terms were associated with the GO term "inflammatory response" (GO:0006954) and 37 HPO terms were associated with the GO term "pigmentation" (GO:0043473). The comparison of manual associations with HPO2GO mappings have revealed that nearly 40% of the HPO terms in the manual associations were also mapped to at least one GO term in HPO2GO. When the same calculation was done to reveal how many of the HPO terms in HPO2GO were also in the manual associations, the result was only 5%. The same correspondence results for GO terms were 35% and 3%, respectively. Finally, the correspondence between the actual HPO-GO mappings has been calculated. Only 23 out of 489 manual associations were retrieved by HPO2GO. An inspection was done to reveal the possible reasons behind the low correspondence, and it was found that, the low number of gene annotations of either the corresponding HPO term or the GO term was the main reason. However, in many cases, an ancestor of the corresponding HPO or GO term was able to be mapped. For example, there was a manual association between "epididymitis: the presence of inflammation of the epididymis" (HP:0000031) and "inflammatory response" (GO:0006954). HPO2GO managed to retrieve an association between "epididymitis" (HP:0000031) and "defense response" (GO:0006952), which is the direct parent term (with "is_a" relationship) of "inflammatory response" (GO:0006954). The "inflammatory response" GO term could not be mapped to any HPO term by HPO2GO since this term was not directly annotated to any gene/protein in UniProt-GOA.

## Performance comparison with the state-of-the-art

The test for the comparison with the state-of-the-art had two objectives: (i) measuring the performance of the method on a temporal hold-out dataset to observe the relevance of the proposed approach, and (ii) investigating how the proposed method competes with the best performing methods in the literature. For this, we have re-generated the HPO2GO mappings using the CAFA2 training set, which contained 133,175 annotations between 5,586 HPO terms and 4,418 proteins, from January 2014. Whereas, CAFA2 evaluation set (i.e., the benchmarking set) contained 19,743 annotations between 1,845 HPO terms and 238 proteins (considering only the no-knowledge benchmark samples). The reason behind the presence of low number of annotations (and proteins) in the evaluation set was that, only the HPO annotations produced between the time of the challenge participation deadline and the end of the annotation collection period (a total duration of nearly 8 months) were used to generate the temporal hold-out evaluation set. One important observation about
the benchmark set is that, a few HPO terms dominates the benchmark set (i.e., the most frequently annotated 185 terms own 55% of all annotations). Also, most of these HPO terms were generic (e.g., HP:0000707 - Abnormality of the nervous system). The small size of the benchmark dataset, together with the uneven distribution of the term frequencies limits the evaluative capacity of this set. All of the datasets, the source code and the supplementary files used in the CAFA2 challenge, and thus in this benchmarking experiment, is available through the CAFA project repositories (URLs: https://github.com/yuxjiang/CAFA2 and https://ndownloader.figshare.com/files/3658395). The CAFA performance evaluation scripts published in these addresses were directly used to calculate the predictive performance of HPO2GO.

HPO2GO mappings generated using the CAFA2 training set contained 27,424 mappings between 2,640 HPO terms and 2,488 GO terms. Considering the whole CAFA2 human target protein set, this mapping produced 1,922,333 HPO predictions for 16,256 proteins and 2,640 HPO terms. The calculated performance of this prediction set was low ($Fmax = 0.30$), mainly due to high number of false positive (FP) hits. However, it is probable that many of these false positives were actually non-documented HPO associations of the corresponding protein, as the benchmark annotation set is incomplete. Increasing the thresholds with the aim of reducing the number of false positives resulted in a matching increase in the number of false negatives (FN), with a similar $Fmax$ value. With the aim of enriching the mappings (to be able to reduce FPs without a significant increase in FNs), HPO annotations of genes from January 2014 (i.e., the CAFA2 training set) were propagated to the root of HPO DAG according to the true path rule. The propagated training set contained 379,513 annotations between 4,418 human proteins and 6,576 HPO terms; as opposed to 133,175 annotations between 4,418 human proteins and 5,586 HPO terms in the asserted CAFA2 set. As observed from the dataset statistics, propagating the annotations have only added about one thousand new terms to the set; however, the number of annotations were significantly increased. Repeating the CAFA2 benchmark analysis using propagated HPO annotations and the same GO annotations set resulted in the same performance ($Fmax = 0.30$). Next, automated GO annotations (i.e., evidence code: IEA) have been included in the source GO annotation set, which increased the number of unique GO annotations from 128,947 to 214,235 (a 66% increase). Using the propagated HPO annotations together with the enlarged GO annotation set, the new HPO-GO mappings, namely "HPOprop2GOall", were generated. The finalized HPOprop2GOall contained 198,928 mappings between 4,780 HPO terms and 5,196 GO terms; as opposed to 27,424 mappings between 2,640 HPO terms and 2,488 GO terms in the original CAFA2 mappings. The drastic difference between the numbers have indicated the enrichment provided by annotation propagation and GO set enlargement. Subsequently, HPOprop2GOall mappings were used to predict HPO associations for all CAFA2 targets, producing 13,022,574 predictions (as opposed to 1,922,333 predictions with the asserted set). Considering only the CAFA2 benchmark proteins, the predictions generated by using the optimized parameters (i.e., $n = 170$ and $S = 0.11$) resulted in 34,486 HPO predictions for 221 benchmark proteins and 235 HPO terms, with a performance of $Fmax = 0.35$ (no-knowledge benchmark sequences in the full evaluation mode), which is among the

top performances considering all of the models from 38 participating groups in the CAFA2 HPO prediction track. The *Fmax* performance of the top model in the challenge was 0.36 (*Jiang et al., 2016*), and the performance of the naïve baseline classifier was also the same. In Fig. 5, each bar displays the overall performance (*Fmax*) of the CAFA2 participators, baseline classifiers and HPO2GO. Additionally, weighted precision–recall curves were plotted, to assess the performance of the method at different threshold selections (Fig. 6). The weighting procedure was done according to the information content of each HPO term; as a result, informative terms received higher weights. The term weights were officially calculated and published by CAFA2 challenge evaluators and these weights were directly used in this study. As shown by the black curve in Fig. 6, HPO2GO performed the same as the best methods on the optimal point (*wFmax* = 0.29), indicated by the circular marking on the curve. Furthermore, the minimum semantic distance (*Smin*) was calculated for HPO2GO and compared again with the CAFA2 participators (Fig. 7). Here, lower *Smin* values indicate higher performance. As shown in Fig. 7, HPO2GO performed slightly worse (*Smin* = 57.2) compared to the top performing methods and the naïve classifier. Finally, term-centric predictive performance of HPO2GO was measured. In the term-centric evaluation, the performance of a predictor is measured independently for each ontology term. Figure 8 displays the average HPO term-centric area under the ROC curve (AUROC) measures for HPO2GO and the CAFA2 participating methods. In this evaluation mode, HPO2GO came second (AUROC = 0.59), with a significantly better result compared to the baseline classifiers. HPO2GO CAFA2 benchmark predictions are available in the repository of the study (filename: "HPO_CAFA2_benchmark_predictions.txt").

The coverage of HPO2GO on the CAFA2 benchmark protein set was nearly 97%. The high coverage indicates that HPO2GO managed to annotate a wide range of proteins. According to the analysis of 6 genes, which codes for the proteins in the CAFA2 benchmark set that HPO2GO could not annotate (gene symbols: *ATOH7*, *DMP4*, *GNT2C*, *CE126*, *PGAP3*, *SERAC1*), only two of these proteins have GO annotations (gene symbols: *ATOH7*, *PGAP3*), and the others had no GO annotations at all with experimental evidence codes; as a result, it was impossible for HPO2GO to assign HPO terms for those proteins. Considering *ATOH7* and *PGAP3*, their annotated GO terms (GO:0003407)—neural retina development and GO:0021554—optic nerve development for *ATOH7*, GO:0016788—hydrolase activity, acting on ester bonds, GO:0006505—GPI anchor metabolic process and GO:0031227—intrinsic component of endoplasmic reticulum membrane for *PGAP3*) have been associated with HPO terms in the raw HPO-GO term mappings; however, all of these mappings were eliminated at the resampling step due to low $n$ and $S$ values.

At this point in the study, CAFA2 benchmark performance test was also repeated using the raw HPO-GO mappings (without statistical resampling) in order to observe the impact of eliminating unreliable term mappings using $n$ and $S$ thresholds. The raw mappings were composed of 879,873 HPO-GO term associations, which led to 66,522,438 predictions between 18,155 proteins and 5,559 HPO terms. The statistics indicate that nearly 65% all possible combinations between the target proteins and HPO terms were produced as predictions. The performance analysis of this prediction set resulted in a very low *Fmax* value (i.e., 0.001), as expected, indicating the effectiveness of the statistical

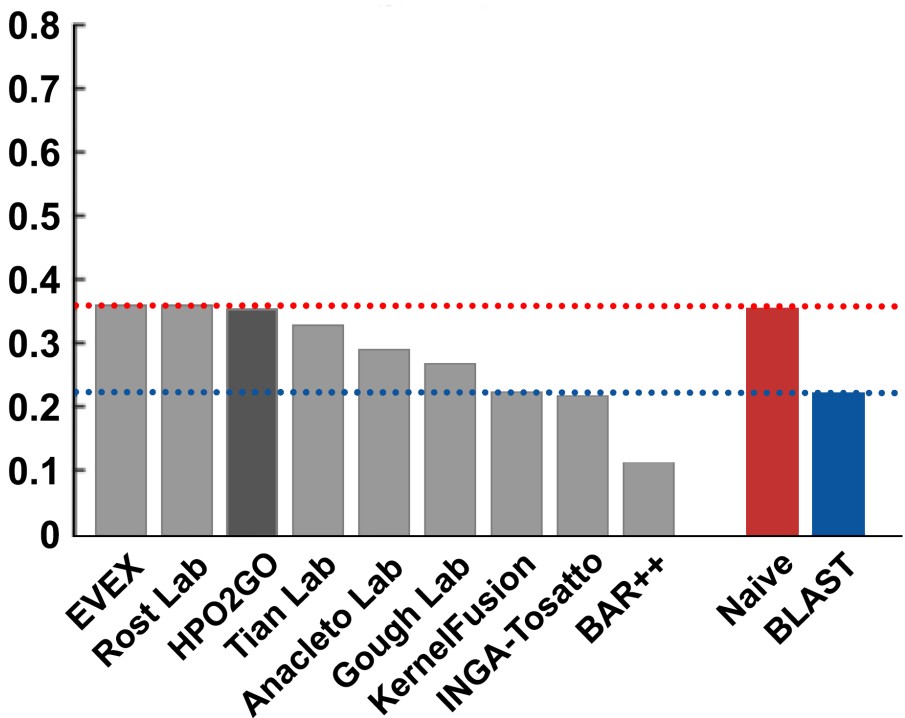

**Figure 5** *Fmax performance results of the CAFA2 HPO prediction challenge.* Performance results (*Fmax*) of the top performing groups (grey bars), baseline classifiers (red and blue bars) and HPO2GO (dark grey bar). The lengths of the bars are directly proportional to the performance.

resampling procedure applied in this study. At all stages of the performance analysis, different HPO2GO mapping sets were generated using various resampling parameters (i.e., different $n$ and $S$ threshold selections), and tested on the CAFA2 benchmark; however, these mappings produced performances slightly inferior to the ones reported above. The most probable reason behind observing reduced performance with a lowered threshold was the inclusion of high number of mappings in the finalized set, most of which were false positives. This in turn provided a reduced precision and a reduced *Fmax* measure. On the other hand, when excessively high thresholds were selected, many of the reliable mappings were probably discarded, leading to high number of false negatives, thus a reduced recall and a reduced *Fmax*, as well.

### Generation of the finalized HPO2protein predictions

Up-to-date HPO2GO mappings were employed to predict HPO terms for the human protein entries in the UniProtKB/Swiss-Prot database (i.e., 20,258 protein records), and the resulting prediction set was marked as the finalized HPO2protein predictions. This set contained 3,468,582 HPO predictions for 18,101 proteins and 3,693 HPO terms. HPO2protein predictions are available in the repository of the study (filename: "HPO2protein_Predictions.txt").

Finally, up-to-date HPO2GO model was run on the CAFA3 human protein targets, which produced 3,453,130 predictions on 16,609 human proteins with 3,719 HPO
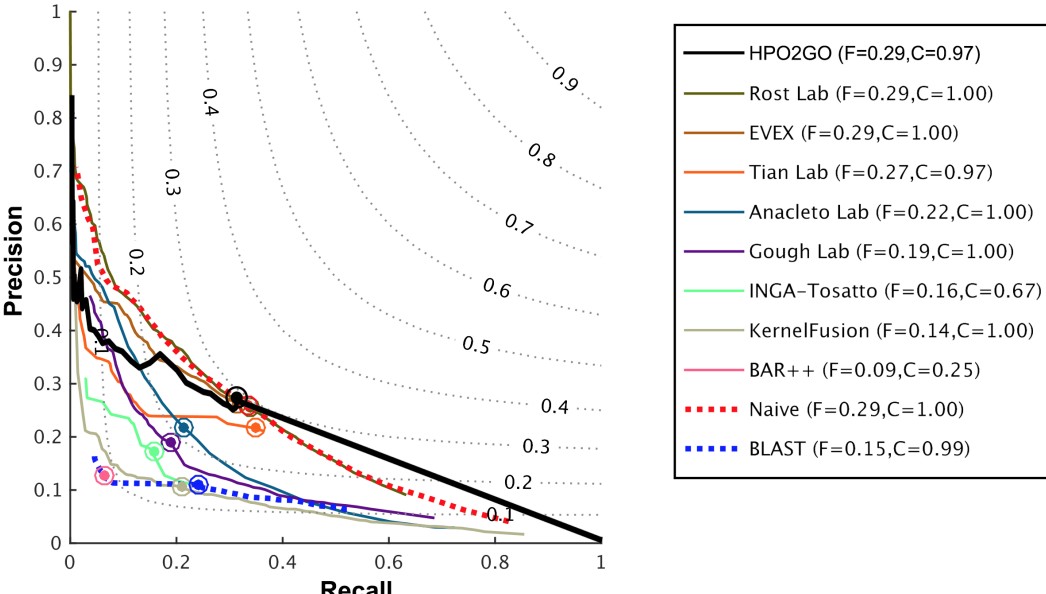

**Figure 6** **Weighted precision–recall curves of the CAFA2 HPO prediction challenge.** Weighted precision–recall curves of the top performing groups (coloured curves), baseline classifiers (red and blue dashed curves) and HPO2GO (black curve). The circle on each curve represent the performance at the optimal threshold. Dashed grey iso-performance curves project the weighted *Fmax*. F (inside the box, to the right side) display the weighted *Fmax* values.

terms. A more stringent subset of this prediction set (i.e., predictions produced from mappings with $S \geq 0.2$) has been officially submitted to the CAFA3 challenge. HPO2GO CAFA3 target predictions are available in the repository of the study (filename: "HPO_CAFA3_target_predictions.txt"). There was a small difference between the number of query proteins in HPO2protein and the CAFA3 target sets (20,258 as opposed to 20,197, respectively). At the time of writing this manuscript, the CAFA3 challenge results have not been announced yet.

## The biological relevance of the selected HPO2GO mappings—a case study

In order to discuss the biological relevance of HPO2GO mappings, selected HPO-GO term mappings were examined. For this purpose, three confidence bins (high-level, mid-level and low-level reliabilities) were determined, considering their respective $S$ and $n$ values (high $S$ and $n$ values together indicate elevated reliability). Six example mappings (two from each confidence bin) were randomly selected for the case study. The first case was a highly reliable mapping between the phenotypic abnormality HPO term "absence of bactericidal oxidative respiratory burst in phagocytes" (HP:0002723) and the GO term "respiratory burst" (GO:0045730), which is in the BP category. The exact definition of this GO term in the UniProt-GOA database is: "*A phase of elevated metabolic activity, during which oxygen consumption increases; this leads to the production, by an NADH dependent system, of hydrogen peroxide (H2O2), superoxide anions and hydroxyl radicals.*" (URL:

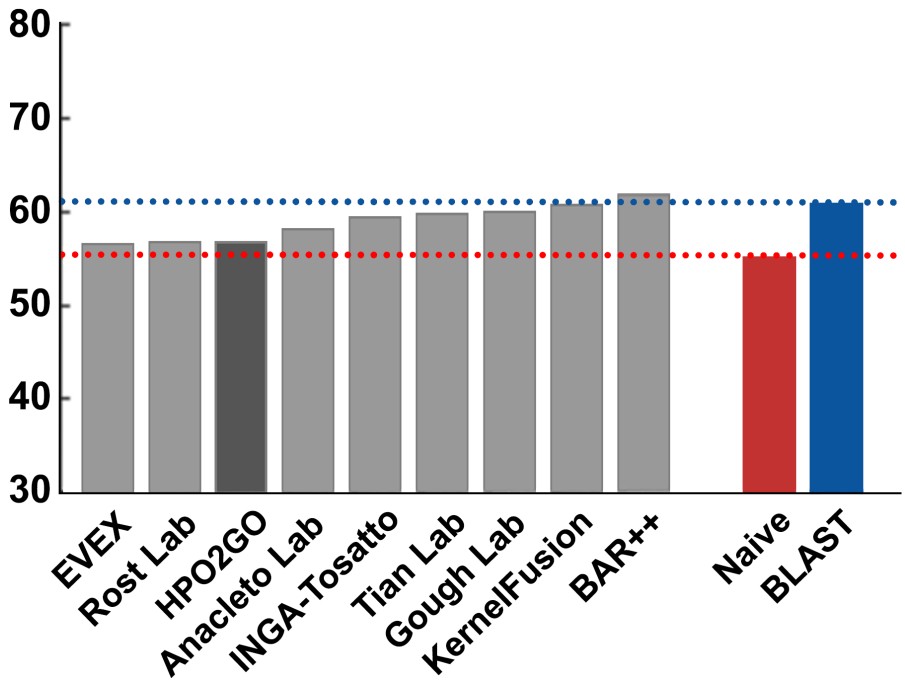

**Figure 7** *Smin* **performance results of the CAFA2 HPO prediction challenge.** Minimum semantic distance performance results (*Smin*) of the top performing groups (grey bars), baseline classifiers (red and blue bars) and HPO2GO (dark grey bar). The lengths of the bars are inversely proportional to the performance.

https://www.ebi.ac.uk/QuickGO/term/GO:0045730) These two terms were mapped to each other in HPO2GO with high confidence (i.e., $S = 0.89$ and $n = 4$). The symbols of the co-annotated genes were *CYBA*, *CYBB*, *NCF2* and *NCF1*. As observed from the names of both terms and from the description of the GO term, the HPO term defines an abnormal condition that corresponds to the absence of the biological process portrayed by the mapped GO term. This is in accordance with the logic behind mapping HPO terms with GO terms, which stated the occurrence of an abnormality (i.e., the HPO term) due to the loss of the biological function defined by the mapped GO term. In addition, there is a GO term named "respiratory burst after phagocytosis" (GO:0045728), which is related (i.e., is_a relationship) to GO:0045730 as its child (descendant) term. These two terms are two-step away from each other on the GO DAG. This term (GO:0045728) defines a more specific function that is the exact opposite of the mapped HPO term (HP:0002723), semantically. Also, there is an evidence for the relation between HP:0002723 and GO:0045728 in both the OBO and OWL formatted ontology files of HPO (URL: http://purl.obolibrary.org/obo/hp.obo and http://purl.obolibrary.org/obo/hp.owl). However, in HPO2GO, GO:0045728 could not be mapped to HP:0002723 due to low coverage in the source GO annotation set. GO:0045728 was only annotated to one gene (symbol: *HCK*), which was not annotated to HP:0002723, as a result, the mapping could not be generated. Nevertheless, the mapped GO term (GO:0045730) still defined a sufficiently related function.

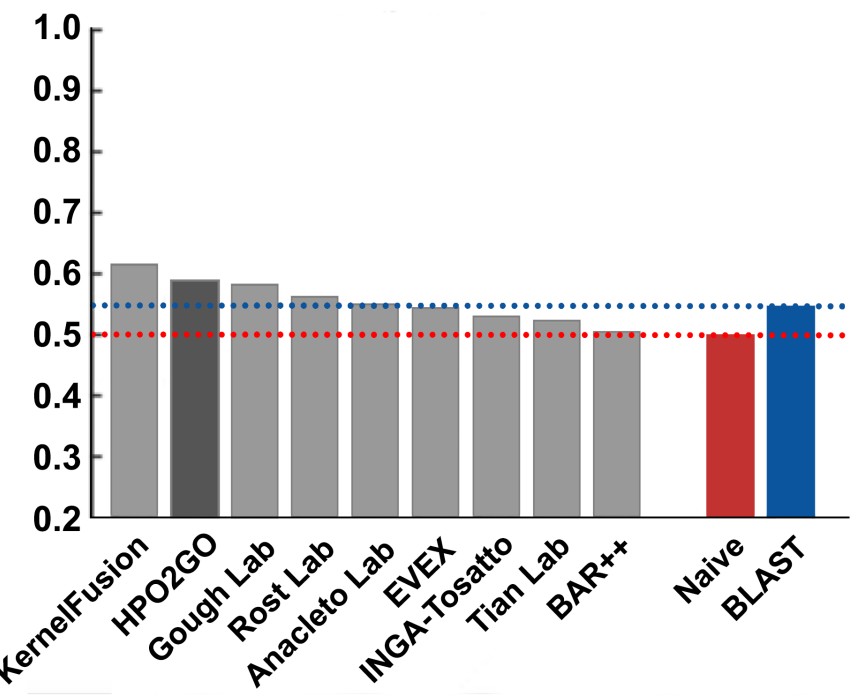

**Figure 8** **Term-centric AUROC performance results of the CAFA2 HPO prediction challenge.** Average HPO term-centric performance results (AUROC) of the top performing groups (grey bars), baseline classifiers (red and blue bars) and HPO2GO (dark grey bar). The lengths of the bars are directly proportional to the performance.

The second selected case with high-level reliability was the mapping between the HPO term "cerebellar hemisphere hypoplasia" (HP:0100307) and the MF category GO term "tRNA-intron endonuclease activity" (GO:0000213) . The exact definition of this specific GO term in the UniProt-GOA database is: "*Catalysis of the endonucleolytic cleavage of pre-tRNA, producing 5′-hydroxyl and 2′,3′-cyclic phosphate termini, and specifically removing the intron*" (URL: https://www.ebi.ac.uk/QuickGO/term/GO:0000213) These two terms were mapped to each other in HPO2GO with high confidence (i.e., $S = 0.86$ and $n = 3$). The symbols of the co-annotated genes were *TSEN2*, *TSEN34* and *TSEN54*. The HPO term HP:0100307 is associated with the disease entry "Pontocerebellar Hypoplasia, Type 2C (PCH2C)" (OMIM:612390) in the OMIM database. According to the disease definition, pontocerebellar hypoplasia is a heterogeneous group of neurodegenerative disorders associated with abnormally small cerebellum and brainstem, and the type 2C is characterized by a progressive microcephaly from child birth (*Barth, 1993*). The occurrence of the disease is associated with missense mutations in either *TSEN2*, *TSEN34* or *TSEN54* genes, which are parts of the tRNA splicing endonuclease complex (*Budde et al., 2008*). It was reported that, due to the abovementioned mutations, there was a partial loss in the function of cleaving the pre-tRNAs by the endonuclease complex (*Budde et al., 2008*). This is another clear example for a HPO term defining an abnormal condition, that is caused by the perturbation in the function defined by the mapped GO term.

An example with mid-level reliability (with parameter values: $0.7 > S > 0.5$ and $10 > n > 2$), where HPO2GO produced a mapping with reduced biologically relevance is the case where the "protein binding" (GO:0005515) MF GO term was associated with the "cognitive impairment" (HP:0100543) phenotype term. The parameter values for this mapping were $S = 0.38$ and $n = 571$. It might be possible to find an indirect connection between some of the diseases that cause cognitive impairment and the loss of protein binding function; however, it is impossible to conclude that all protein binding function losses would result in cognitive impairment. Both of these terms are quite generic; as a result, the mapping is not informative. The interesting observation here is the extremely high $n$ value, which is resulted from the fact that protein binding GO term is a generic and frequent term that is annotated to many different protein entries. This result also indicated that using the $n$ value alone to judge the reliability of a mapping would not be sufficient, instead both $S$ and $n$ should be taken into account.

Another example with mid-level reliability, that was randomly selected from the mappings, was the association between the HPO term "anemic pallor" (HP:0001017) and the CC category GO term "Fanconi anaemia nuclear complex" (GO:0043240). The exact definition of this GO term in the UniProt-GOA database is: "*A protein complex composed of the Fanconi anaemia (FA) proteins including A, C, E, G and F (FANCA-F). Functions in the activation of the downstream protein FANCD2 by monoubiquitylation, and is essential for protection against chromosome breakage*" (URL: https://www.ebi.ac.uk/QuickGO/term/GO:0043240). The textual definition of the corresponding HPO term (HP:0001017) is "*A type of pallor that is secondary to the presence of anemia*". These two terms were mapped to each other in HPO2GO with mid-level confidence (i.e., $S = 0.64$ and $n = 8$). The symbols of the co-annotated genes were *FANCA, FANCB, FANCC, FANCE, FANCF, FANCG, FANCL* and *FANCM*. These genes are parts of the Fanconi anaemia (FA) protein complex, which protects the cell against chromosomal breakage (*Pace et al., 2002*). The corresponding CC GO term (GO:0043240) directly describes the FA complex; whereas, the mapped HPO term (HP:0001017) is associated with the Fanconi anemia disease sub-types in the HPO database (e.g., OMIM:600901, OMIM:227650, OMIM:227645, OMIM:227646). According to the disease definition in OMIM, Fanconi anemia is a heterogeneous disorder associated with genomic instability, mainly characterized by developmental abnormalities in major organ systems (*Deakyne & Mazin, 2011*). Each of the different sub-types listed in OMIM is associated with a mutation in a different FA complex gene. This is a clear example for a HPO term defining an abnormal condition, that is caused by the disease rooted from the disfunction in the biomolecular complex defined by the mapped GO term.

An example with low-level reliability (parameter values: $0.2 > S > 0.1$ and $n = 2$) is the case where "abnormality of reproductive system physiology" (HP:0000080) HPO term was associated with "collagen catabolic process" (GO:0030574) BP GO term. The parameter values for this mapping were $S = 0.11$ and $n = 2$. The symbols of the co-annotated genes were *COL7A1* and *MMP1*. The definition of this GO term in the UniProt-GOA database is: "*The proteolytic chemical reactions and pathways resulting in the breakdown of collagen in the extracellular matrix, usually carried out by proteases secreted by nearby cells.*" (URL: https://www.ebi.ac.uk/QuickGO/term/GO:0030574). On the other hand, HP:0000080 is a

generic phenotype term that has 45 descendent terms, and it is possible to reach the root of the sub-ontology from this term in just three term-to-term jumps. Naturally, HP:0000080 has been associated with high number of diseases (i.e., 465) in the HPO database, one of which is the Myotonic dystrophy (OMIM:160900). Myotonic dystrophy is an autosomal disorder characterized by muscular dystrophy, myotonia, hypogonadism (i.e., functional activity related issues in the testes/ovaries), and etc. (*Musova et al., 2009*). In this sense, the relationship between muscular dystrophy and the collagen breakdown is evident; as a result, it can be stated that HPO2GO identified a relevant mapping. However, HP:0000080 is quite generic and associated with many other processes besides muscular dystrophy; consequently, the target mapping is not very specific.

The last randomly selected case study example is another one from the low-level reliability bin. The HPO term "polycythemia" (HP:0001901) and the BP category GO term "bicarbonate transport" (GO:0015701) were mapped to each other with $S = 0.15$ and $n = 2$. The symbols of the co-annotated genes were *HBA1* and *HBB* (i.e., haemoglobin sub-units). The definition of GO:0015701 in the UniProt-GOA database is "*The directed movement of bicarbonate into, out of or within a cell, or between cells, by means of some agent such as a transporter or pore*" (URL: https://www.ebi.ac.uk/QuickGO/term/GO:0015701). One of the ways the carbon dioxide is removed from tissues is first the generation of carbonic acid, and then the decomposition of carbonic acid into bicarbonate to be transported to the lungs, via the red blood cells (i.e., erythrocytes). This process also drives the transport of oxygen molecules from blood to the tissues via an allosteric mechanism. Thus, bicarbonate transport process have high importance for oxygenation. The definition of HP:0001901 in the HPO database is "*Polycythemia is diagnosed if the red blood cell count, the haemoglobin level, and the red blood cell volume all exceed the upper limits of normal*". HP:0001901 is also associated with 23 diseases most of which are blood related. A few of these diseases are polycythemia vera—PV (OMIM:263300) and various sub-types of familial erythrocytosis (OMIM:13310, OMIM:263400, OMIM:609820, OMIM:611783). Erythrocytosis is an autosomal disorder characterized by increased haemoglobin concentration, increased mass of serum red blood cells, and etc. (*Kralovics, Sokol & Prchal, 1998*). It is probable that certain mutations in the proteins that take part in the bicarbonate transport process would cause a disruption in this function, which would in turn lead to decreased efficiency in both the oxygenation of tissues, and the removal of carbon dioxide from them, causing the polycythemia phenotype related diseases such as polycythemia vera or erythrocytosis. In this example, HPO2GO managed to return an indirect but relevant association.

## DISCUSSION

HPO project's manual HPO-GO term associations have been generated only for a sub-set of HPO and GO terms, by comparing the term definitions. As a result, the coverage of these associations is limited (i.e., a total of 489 mappings). In HPO2GO approach, all GO-HPO term combinations that satisfy the co-occurrence similarity test conditions were linked. This way, the non-documented relations were also identified. The results of the mapping comparison analysis has indicated that the direct correspondence between the manual

associations and HPO2GO was low, mainly due to the limitations in the source annotation sets. Nearly all of the manual HPO-GO associations had highly similar definitions. Whereas, HPO2GO retrieved associations with mostly dissimilar term definitions, which often made biological sense considering the underlying molecular mechanisms. The manual identification of this type of associations may require a comprehensive curation process. In this context, it is expected that the HPO2GO mappings will be valuable for the research community.

In this study, individual terms from both ontologies were mapped to each other considering the co-annotated genes/proteins. However, the initial design of the experiment considered the mapping of an HPO term to a trio of GO terms, one from each GO category (i.e., biological process—BP, molecular function—MF and cellular component—CC). This way, the corresponding phenotypic abnormality would be associated with a problem in a specific molecular event (defined by the MF term), as a part of a defined large-scale process (the BP term), occurring at a particular sub-cellular location (the CC term). This approach would have been biologically more relevant compared to the current design; however, the initial design failed due to the scarcity of both HPO annotations and MF, BP and CC GO term containing triple annotations. At this point in the study, a second option was considered, where HPO terms were tried to be mapped to MF and BP term pairs; nevertheless, the same problem was encountered again. Reliable annotation sets with higher coverage, which may become available in the future with more curation efforts, may solve this problem and make the abovementioned mapping approach practical. However today, even for the currently applied one to one term mapping approach, the main challenge is the low coverage of the predicted associations due to the small size of the source annotation sets. There can be a few alternative solutions to this problem. First of all, the training sets with enriched GO annotation may be obtained by including the annotations with evidence codes of reduced reliability (e.g., IEA—electronically generated), as this approach has been shown to work well with the CAFA2 benchmark set. Another option for enlarging the GO annotations would be utilizing the gene/protein similarity information (i.e., protein function prediction). Scaling up the coverage of both the HPO and the GO sets can be provided by propagating the annotations to the parent terms according to the true path rule. "HPO2GO asserted vs. propagated mappings" analysis have indicated that this approach significantly increases the coverage. Another option here would be assuming a more elaborate approach in the mapping procedure by taking the graph-based hierarchical term relationships into account while generating the HPO2GO mappings (i.e., the parent and child terms of the target HPO-GO term pair, that are co-annotated to different genes/proteins, will also contribute to the calculation of the co-occurrence similarity of the target HPO-GO pair). This approach have been widely accepted in the area of sematic similarity based functional analysis of biomolecules.

The official CAFA2 challenge results have indicated that, the methods based on sequence similarities (e.g., the baseline classifier BLAST and a few models from the participating groups) can achieve a good predictive performance considering the GO terms in the molecular function (MF) category. This was expected since it is possible to detect most of the signatures related to the molecular functions by analysing the amino acid sequence.

However, most of the sequence-similarity based methods failed in predicting the cellular component (CC) GO term and HPO term associations. This can be explained for CC terms as either by the cleavage of the signals from the sequence post-translationally or the difficulties in detecting weak signals used for directing proteins to different compartments. Considering the HPO prediction, the case may completely be different. As opposed to GO terms, which define the attributes the proteins contain, HPO terms define phenotypic abnormalities caused by the protein when it loses one (or more) of its functions, usually due to certain mutations in the gene that codes the protein. Due to this reason, transferring a HPO annotation from one protein to another based on sequence similarity does not have a biological relevance, which explains the poor performance of the BLAST classifier.

It was interesting to see that the HPO2GO CAFA2 benchmark set predictions produced using the training set of asserted annotations resulted in the same overall performance as the predictions produced using the training set of propagated annotations (without the inclusion of electronically made GO annotations). In theory, employing the true path rule propagation would enrich the training set (i.e., less false negatives) without any sacrifice in terms of the type I error (i.e., the same number of false positives), which should have produced an elevated *Fmax* value. Instead, the performance remained the same. There could be two possible reasons for this. First, there were already false positive instances in the training set beforehand, and the propagation process just made the situation worse by increasing the number of false positive instances in a magnitude equal to the number of ancestor terms of the false positive instances (i.e., equal to the number of propagation operations). This in turn compensated for the increase in the coverage obtained by propagating the annotations. The second reason could be that, there were no significant errors regarding the propagated HPO2GO predictions; however, the benchmark annotations were incomplete. As a result, a portion of the real true positive predictions were counted as false positives. Nevertheless, employing the propagated HPO annotations together with the enlarged GO annotation set (including the electronically made annotations) have increased the predictive performance by nearly 17%. However, it is not possible to be sure about the accuracy of these results due to the small size of the CAFA2 test dataset. A larger benchmark annotation set that is guaranteed to be complete would be required in order to discuss the performance further.

An important observation regarding the CAFA tests done in this study is that, there was a large difference between the number of HPO predictions for CAFA2 and CAFA3 targets, using HPO2GO with default parameters (i.e., 1,922,333 in CAFA2 as opposed to 3,453,130 in CAFA3). There was also an increase in the number of predicted HPO terms (i.e., 2,640 in CAFA2 as opposed to 3,719 in CAFA3), and there were no significant increase in the number of targets. The increase in the number of predictions and the predicted HPO terms can be attributed to the training set getting larger and more informative in time. The training set used for CAFA2 contained 133,175 annotations; whereas, it was 153,575 for CAFA3. The comparison of the predictive performances of HPO2GO trained by the CAFA2 and the CAFA3 training sets may reveal more about the situation.

Considering the HPO2GO biological relevance case studies, it was possible to find a biological connection between the mapped HPO and GO terms in most cases; however,

the connection became indirect and more abstract, when the $S$ and $n$ values were low (i.e., reduced reliability). Frequently, highly generic/shallow HPO and GO terms were mapped to multiple terms from the other ontology, with generally low parameter values. These mappings were still observed to be relevant but less informative. On the other hand, most of the mappings with high-level reliability were between specific HPO and GO terms, and the selected cases from these mappings were observed to be highly informative. Similar to the other automated methods that produce predictions on the biological data, HPO2GO has limitations. This was reflected in the results of the predictive performance test on the CAFA2 benchmark set ($Fmax = 0.35$), which can be considered low for real-life applications. For HPO2GO to be employed in biological data analysis pipelines in the future, HPO-GO term mappings should also be manually curated.

## CONCLUSION

In this study, a simple and effective strategy, HPO2GO, was proposed to semantically map phenotypic abnormality defining HPO terms with biomolecular function defining GO terms, considering the cross-ontology annotation co-occurrences on different genes/proteins. This approach can easily be translated into novel HPO term predictions for genes/proteins. A literature based case study was carried to discuss the biological relevance of the selected HPO2GO mappings. This work also presents an application of the cross-ontology term mapping approach by generating HPO-protein associations. HPO2GO was benchmarked on CAFA2 challenge protein targets and it was revealed that the method was among the best performers of the HPO term prediction track participators (i.e., the state-of-the-art methods). Also, the up-to-date trained system was employed to predict HPO associations for all human proteins in the UniProtKB/Swiss-Prot database (i.e., HPO2protein predictions). The methodology proposed here may also support the already established approaches (e.g., text mining), as it is possible for different techniques with different data sources and perspectives to produce results that complement distinct missing pieces of the knowledge space. This property is often utilized in ensemble based classification approaches. In this sense, it would also be interesting to analyse the complementarity between the predictions of the proposed method and the predictions of the state-of-the-art approaches participated in CAFA2 challenge; however, this was not possible since the actual prediction results of the participant groups are not publicly available.

As for the future work, it is first planned to map the HPO terms to GO term trios (i.e., MF, BP and CC terms at the same time) using enriched annotation datasets, as explained in the discussion section. Another future task is the integration of HPO2GO mappings to our freely available GO based automated protein function prediction tool/server UniGOPred (*Rifaioglu et al., 2018*) so that query proteins that receive a GO term prediction will be automatically associated with the HPO term(s) that are mapped to the corresponding GO term. It is expected that this approach would produce large-scale HPO predictions for uncharacterized proteins without any curated annotation, where the only available information is the amino acid sequence. The knowledge extraction methodology proposed here can easily be combined with various types of protein features employed in other

predictive methods (e.g., variant information, PPIs, gene expression profiles, etc.) to generate an ensemble HPO term prediction tool that identifies novel HPO-gene/protein-disease associations.

### Funding

The authors received no funding for this work.

### Competing Interests

The authors declare there are no competing interests.

### Author Contributions

- Tunca Doğan conceived and designed the experiments, performed the experiments, analyzed the data, contributed reagents/materials/analysis tools, prepared figures and/or tables, authored or reviewed drafts of the paper, approved the final draft.

### Data Availability

GitHub: https://github.com/cansyl/HPO2GO.

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
