# Peer review of "HPO2GO: prediction of human phenotype ontology term associations for proteins using cross ontology annotation co-occurrences"

_PeerJ, doi:10.7717/peerj.5298_

## Round 0.1 · original submission · Major Revisions

Your paper has been reviewed, and a number of questions and problems were raised which will require a major revision to the manuscript. Please respond to all of these carefully if you decide to resubmit.

In particular, please address the concerns raised by reviewer 1
regarding the validity of your experiments on the CAFA2 dataset, and
the concerns expressed by reviewer 2 on the biological relevance of
the predictions and only using asserted annotations rather than
employing the true-path rule.

·

Basic reporting

The paper is well structured. Has a good introduction and a background. References are adequate for the most part. But the writing is not clear at times (some sentences are too long etc.). It is suggested that the writing be reviewed by a native English speaker for improving the readability. Figures are relevant, high quality, well labelled & described (except for Fig 5, which is of poor quality). Raw data are accessible and reasonably well documented. Following is the list of all issues (starting with their line numbers).

Major comments:
Introduction: missing description about sub-ontologies of HPO. Mention the DAG structure of HPO. Are there evidence codes in HPO?
Method, Dataset construction: Which sub-ontologies of HPO are used for your work?

Minor comments:

18: change “provided” to provides
35: change “beat” to beats
46: change “and etc.” to etc.
50: change “source of” to sources of
81: change “basic” to a better word
96: change “with annotation” to with the annotation
103: change “CAFA organization have” to has
121-124: Sentence is too long and does not make sense. Please reword.
133: change “approach” to approaches
134: change “coverage of especially” to coverage, especially
143: change “consist: to consisting
143: define PPIs
155-156: this sentence is not true. RANKS, PHENOStrcut, dcGO are not among the top performers of CAFA2. In fact, RANKS and PHENOstrct did not participate in CAFA2 (Jiang et al., 2016).
164: citation is required to a study showing this bias
165: “concept” may need a better word for this as the meaning of the sentence is not clear
187: change “as for example” to for example
188: “problematic version” requires a different phrase to improve meaning
193-194: “The function usually..” - how do you know for sure?
201: “time-held” may need a different word since this word is not used often
220-221: this sentence suggests that ancestor terms are not reliable – which is not true. May need to reword.
237: change “whole” to complete
243: change “probable to for” to “probable for”
244: “randomly occurred mapping” may need to be replaced by better wording
245-247: need a citation for the sentence starting with “For each ..”. It looks like your equation (1) is based on information gain.
300: change “participate to” to “participate in”
320-324: Define Tau in equations 2, 3 and 4
335: change “obtains” to obtain
365: change “displays” to display
368: change “leaves” to leave
370: change “shows” to show
374: period instead of comma at the end of sentence
431: Wrong name of GO term used for GO:0045728
432: “GO:0045728 which is related to the mapped term GO:0045730 on the GO DAG:. Explain how these two terms are related and specify the relation type.
503-507: “prediction results are available” on 503 and “results not been announced “ on 506 conflicts with each other. Also, “small difference” does not make sense. Elaborate more.
513: “quite limited”: provide evidence
Fig 5 is of low quality – reproduce.

Experimental design

Clearly specify how “genes_to_phenotypes” were created by the HPO curators. Isn’t it possible that some of the same resources/information used for adding GO annotations is used for this (by HPO curators)? If so, your experiments are biased. If not, clarify.

312: It is stated that for the CAFA2 comparison, HPO2GO mappings were re-generated with training data. Does that mean you used both GO and HPO from 2014 Jan?

Following three reasons invalidate the CAFA2 comparison results (Figures 5 and 6). This reviewer feels very strongly about correcting these issues before accepting the paper (because Figs 5 and 6 provide the primary evidence that HPO2GO is effective).

1. 464: Do you use GO annotations from the same period? If not, the results are invalid because you are using 3 years’ worth of GO annotations that were inserted to GO database after the submission deadline (for which the other teams did not have access, regardless of whether they used GO as input or not)

2. Also, it says the CAFA2 training data is from Oct 2015. According to CAFA2 paper (Jiang et al., 2016), they are from Sep 2013. If this is the case, this is another reason that invalidates results.

3. 476: Clearly mention that you are using “No knowledge” mode. Are you using the exact same evaluation method used in CAFA2 that uses bootstrapping (Jiang et al., 2016). If not, results are invalid. If you did use the same method, clarify.

365: states that you used different n values in Figure 3. Explain why you chose n=1, 5, 25, and 75 in Figure 3. Also, there is another explanation in line 398 that these values “indicate the distribution differences at extreme cases”. Explain how these cases are extreme.

416: in section “The biological relevance of the selected HPO2GO mappings - A case study”, you have chosen only two different examples. Explain why you chose these two examples. E.g. was it because of the high S value?

Explain why you did not use TAS and IC codes for GO annotations (they are used in CAFA2 evaluation)

You should use other measures in addition to Fmax too (Smin and term-centric measures). According to CAFA2 paper (Jiang et al., 2016), there is large variance in ranking of methods when using different measure (for HPO).

Validity of the findings

35, 482: Due to the issues in experimental design mentioned above, your claim that HPO2GO “beats all other top performers” cannot be considered valid.

573-575: “This approach can easily be translated into novel HPO term predictions for genes/proteins, as well as into new HPO-disease or gene-disease associations.”: Not enough evidence provided in the paper to make the second part of the claim.

582-583: “The methodology proposed here was only meant to support the already established approaches (e.g., text mining), since different techniques with different data sources and perspectives produce results that complement distinct missing pieces of the knowledge space.”: not enough evidence provided in the paper to make this claim (no comprehensive experiment /study for making this claim).

Additional comments

It would be nice to see an analysis of which proteins were missing predictions from your method (evidenced by C = 0.97) and provide insights/ reasoning on why.

It would be interesting to mention how many methods (you compare to) use GO as input.

It would be Interesting to see a comparison of performance between raw mappings vs finalized. This should highlight the importance/ impact of your sampling method.

489: “however, these mappings produced performances slightly inferior to the one generated using the optimal thresholds”. It would be helpful to have a discussion on this observation and why this happened.

·

Basic reporting

This manuscript describes an approach to generate cross-ontology
mappings from HPO to GO in order to propagate GO annotations to HPO
annotations, increasing coverage of HPO annotations. The mappings are
created by first using a Jaccard-like measure based on shared genes
between GO and HPO gene sets. Threshold filters are determined using
randomized data.

The approach shows better performance on CAFA2 HPO prediction tests
measures than all other methods. However, as to be expected with any
guilt-by-association method the overall accuracy is low, which limits
the applicability of the approach in real-world applications.

The approach is clearly explained, the author does a good job of
presenting background material. The fact the approach outperformed
existing CAFA2 methods is commendable. However, there needs to be
better biological evaluation of the mappings. Additionally, further
experiments with different scoring measures and annotatuon propagation
would benefit the work greatly, but are not strictly required so long
as the limitations are clearly explained.

Experimental design

My comments are ordered from most important to least important

1. The evaluation of biological relevance is incomplete.

The author performs two evaluations, one using CAFA2 and the other by
examining selected mappings and comparing against the literature. The
The first evaluation is objective and clear, the second evaluation is
problematic.

First, only two mappings are selected and it's not clear if these were
cherry picked. Given there are ~46k mappings it's relatively easy to
pick a handful that have a good story.

The first example has a mistake that should be fixed. The match for
"absence of bactericidal oxidative respiratory burst in phagocytes" is
actually to the less precise "respiratory burst" than "respiratory
burst after phagocytosis" as the authors claim.

But the real problem is that if a set of mappings were selected at
random and then evaluated it would reveal that many of the mappings
are for pairs terms that don't appear to be related in any way, and
genes are shared by chance.

For example, the following two mappings have relatively high scores,
in the top 3% of the complete set of mappings:

HP:0000006-Autosomal dominant inheritance GO:0005515-protein binding 0.5850
HP:0000007-Autosomal recessive inheritance GO:0005515-protein binding 0.5734

It's hard to see any biological justification here, and both HPO
inheritance terms and protein binding appear multiple times. If these
were to be used as a basis for suggesting new annotations the false
positive rate would be too high for this to be useful.

To rectify this: minimally, an ubiased sample of mappings should be
randomly selected and evaluated. This could be done from different
score bins.

Additional experiments would be to see how well HPO2GO augmented
annotations would perform on tasks the HPO is designed for such as
disease diagnosis.

2. The decision to use only asserted annotations is not well justified

The manuscript states:

220 as well. In this study, the asserted annotations are used in the analysis (in terms of both GO and
221 HPO), in order to make sure the training set includes only the most reliable annotations.

It is not true that logically propagating annotations up to the root
would have produced less reliable annotations. In fact these are
logically guaranteed to be correct. Eliminating this would reduce
recall with no gain in precision.

I recommend that the experiment is redone using propagated
annotations.

Failing this, the manuscript should explicitly state the lack of
propagation as a limitation, or provide a valid reason for excluding
them.

3. The decision only to use experimental GO evidence is not well
justified

This is related to the previous point. The previous point concerned
logical propagation up the hierarchy which is always guaranteed to be
valid, for HPO or GO. This point pertains to the propagation of GO
annotations from other species. In this case, propagation is not
guaranteed to be valid (due to species difference, natural evolution,
etc). However, the complete set of GO annotations are generally
regarded to be a good default choice, evidence codes such as ISS and
IBA are high quality, and methods such as propagating based on
interpro2go are reliable.

I think the results would potentially have been better had these been
included. Ideally there would be an experiment comparing the two.

I recommend including these, or alternatively more clearly stating
that this is a limitation of the approach, or providing clearer
justification for the current choice.

4. Comparison with logical definitions

The HPO includes manually curated OWL definitions in which HPO classes
are connected to GO, Anatomy ontologies, etc. An obvious test is to
validate the predicted HPO2GO against this manual one.

The manuscript states:

516 community. It would also be interesting to compare the HPO2GO mappings with the
517 abovementioned manually curated associations; however, it is not possible to access this data in
518 the HPO repository anymore.

In fact these are now part of the official OWL version of HPO.

5. Choice of scoring

The scoring method (Eq1) is similar to Jaccard, but also uses a
statistical procedure to see thresholds of number of genes and
scores.

It's not clear why a straightforward fisher exact test is not used or
compared against. This has the advantage of providing a p-value for
each mapping indicating the probability that such a mapping would be
observed given the null hypothesis. Standard multiple hypothesis
corrections methods can be used.

It may be the case that Eq1 has some advantages but additional scoring
methods should have been explored.

Validity of the findings

See section 2

Additional comments

The discussion section is a good overview of some of the challenges,
and demonstrates the author has understood some of the nuances of
predicting HPO annotations, and why simple approaches like BLAST would
have low precision.

e.g.

557 Due to this reason, transferring
558 a HPO annotation from one protein to another based on sequence similarity does not have a
559 biological relevance, which explains the poor performance of the BLAST classifier.

This is a great suggestion:

589 As for the future work, it is first planned to map the HPO terms to GO term trios (i.e., MF, BP and
590 CC terms at the same time) using enriched annotation datasets, as explained at the Discussion
591 section

---

## Round 0.2 · accepted · Accept

The detailed care with which you addressed reviewer comments is appreciated!

# ·

Basic reporting

No comment.

Experimental design

No comment.

Validity of the findings

No comment.

Additional comments

Thank you for carefully addressing all my comments.